# CROSS-CORPUS TRAINING WITH TREELSTM FOR THE EXTRACTION OF BIOMEDICAL RELATIONSHIPS FROM TEXT

## ABSTRACT

A bottleneck problem in machine learning-based relationship extraction (RE) algorithms, and particularly of deep learning-based ones, is the availability of training data in the form of annotated corpora. For specific domains, such as biomedicine, the long time and high expertise required for the development of manually annotated corpora explain that most of the existing one are relatively small (*i.e.,* hundreds of sentences). Beside, larger corpora focusing on general or domain-specific relationships (such as citizenship or drug-drug interactions) have been developed. In this paper, we study how large annotated corpora developed for alternative tasks may improve the performances on biomedicine related tasks, for which few annotated resources are available. We experiment two deep learning-based models to extract relationships from biomedical texts with high performance. The first one combine locally extracted features using a Convolutional Neural Network (CNN) model, while the second exploit the syntactic structure of sentences using a Recursive Neural Network (RNN) architecture. Our experiments show that, contrary to the former, the latter benefits from a cross-corpus learning strategy to improve the performance of relationship extraction tasks. Indeed our approach leads to the best published performances for two biomedical RE tasks, and to state-of-the-art results for two other biomedical RE tasks, for which few annotated resources are available (less than 400 manually annotated sentences). This may be particularly impactful in specialized domains in which training resources are scarce, because they would benefit from the training data of other domains for which large annotated corpora does exist.

## 1 INTRODUCTION

Relationship Extraction (RE) from text is a Natural Language Processing (NLP) task that aims at extracting automatically and summarizing in a structured form the unstructured information of texts. A relationships takes the form of a labeled link between two named entities as illustrated in Figure 1. Given two identified entities, the RE extraction task consists in predicting whether their is a relation between them and if so, the type of the relation. It can be seen as a classification task by computing a score for each possible relation type, given a sentence and two identified entities. Deep learning methods have demonstrated good ability for such tasks Zeng et al. (2014), but one of their drawbacks is that they generally require a large amount of training data, *i.e.*, text corpora where entities and relationships between them are annotated, in order to obtain reasonable performances. The building of such a corpus for a specific task, such as those of interest in biomedicine, is time consuming and expensive because it implies complex entities (*e.g.*, genomic variations, complex symptoms), complex relationships (which may be hypothetical, contextualized, negated, n-ary) and requires trained annotators. This explain why only few and relatively small (*i.e.*, few hundreds of sentences) corpora are available, making these resources particularly valuable. Among these tasks, one can mention the extraction of genomic variations-phenotype relationships for which only a manually annotated corpus of 362 sentences, named SNPPhena exists (Bokharaeian et al., 2017). Beside, several larger corpora have been manually annotated with biomedical or general-domain relationships and made available (Hachey et al., 2012; Herrero-Zazo et al., 2013; Gurulingappa et al., 2012). Because these corpora share the same language (*i.e.*, English) and thus a common

The rs165774 SNP was associated with alcohol dependence.

Figure 1: Example of a relation labeled *Weak Confidence association* between two named entities: a *single nucleotide polymorphism (SNP)* and a *phenotype*, from the corpus SNPPhenA.

syntax, we wonder if these resources, developed for slightly different tasks, may be reused for extracting relationships in domain with scarce resources. Several multi-task learning approaches have been proposed to improve performance for a given task using corpus developed for related tasks Collobert et al. (2011). In this paper, we investigate a cross-corpus strategy to improve performances for biomedical RE tasks for which few training data are available, using larger additional corpora developed for other specific RE tasks. This is done by jointly training deep learning-based models while sharing some of the parameters.

Before or beside deep learning methods, other approaches for RE have been proposed. Co-occurrence-based methods for instance assumes that two entities mentioned frequently in the same unit of text (such as a sentence or a paragraph) are related (Garten & Altman, 2009). Rule-based methods use manually designed, or learned, rules consisting of word morphosyntactic features or sentence-level syntactic features (Fundel et al., 2007). These methods have the advantage of requiring few or no annotated data.

Within machine learning methods, deep learning ones enable to model complex structures such as natural language and successfully applied to various NLP tasks. In particular, it as been successfully applied to RE by training from annotated corpora (Zeng et al., 2014) While other methods mainly depend on the quality of extracted features derived from preexisting NLP systems (*e.g.*, POS tagger, stemmer, lemmatizer or syntactic parser), deep learning models automatically learn lexical features using continuous word vector representations, usually named *word embeddings*, and sentence level features using deep neural network such as Convolutional Neural Network (CNN) (LeCun et al., 1998) or Recursive Neural Networks (RNN) (Pollack, 1990).These models achieve good performances, but strongly depend on the existence of large training corpora, which make them difficult to use for tasks associated with scarce resources.

In this paper investigate within four specific RE tasks, for which only few training data are available, how large annotated corpora can be used to improve performances of deep neural networks. We experiment two different deep learning approaches that have been previously used for RE. The first is a Multi-Channel CNN (MCCNN)-based model used in (Quan et al., 2016) for biomedical RE and the second is the tree-structured Long Short Term Memory (TreeLSTM) model (Tai et al., 2015), which have been adapted with success for RE (Miwa & Bansal, 2016). The main difference between these two models is the ability of the latter to exploit the syntax of the language by including a dependency tree structure in the vector representation of sentences.

We conduct our experiments using two relatively small biomedical corpora, SNPPhenA and EU-ADR. Both contains less than 400 manually annotated sentences for each task, but note that EU-ADR focus on three different tasks. As supplementary data, we used three larger corpora: SemEval 2013 DDI, ADE and reACE. Details on these five corpora are provided Section 4. Our experiments show that contrary to the MCCNN model, the TreeLSTM model benefit from a cross-corpus learning strategy to improve the RE performances for tasks associated with scarce resources. This is done by training a model with data from two distinct corpora, one small and one large, while sharing the model parameters. In addition, our approach led to state-of-the-art performances for the four biomedical tasks associated with scarce resources.

Section 2 review various deep learning methods used for RE and previous multi-task learning approaches. Section 3 details the MCCNN and TreeLSTM models we use. Section 4 describes corpora used in this study and Section 5 presents our experiments and results. We then conclude with a short discussion section.

## 2 RELATED WORK

### 2.1 DEEP LEARNING-BASE RELATION EXTRACTION

Deep learning models, based on continuous word representations have been proposed to overcome the problem of sparsity inherent to NLP (Huang & Yates, 2009). In Collobert et al. (2011), the authors proposed an unified CNN architecture to tackle various NLP problems traditionally handle with statistical approaches. They obtained state-of-the-art performances for several tasks, while avoiding the hand design of task specific features. These results led to progress on NLP topics such as machine translation (Cho et al.), question-answering (Bordes et al., 2014) and RE.

In particular, Zeng et al. (2014) showed that CNN models can also be applied to the task of RE. In this study, they learn a vectorial sentence representation, by applying a CNN model over word and word position embeddings. This representation is then used to feed a softmax classifier (Bishop, 2006). To improve the performance of the RE, other authors consider elements of syntax within the embedding provided to the model: Xu et al. (2015) use the path of grammatical dependencies between two entities, which is provided by a dependency parsing; Yang et al. (2016) include the relative positions of words in a dependency tree. They also take dependency based context (*i.e.,* child and parent nodes) into account during the convolution.

Beside CNN models that incorporate syntactic knowledge in their embeddings, other approaches go further by proposing neural networks which topology is adapting to the syntactic structure of the sentence. In particular, RNN have been proposed to adapt to tree structures resulting from constituency parsing (Socher et al., 2013; Legrand & Collobert, 2014). In that vein, Tai et al. (2015) introduced a TreeLSTM, a generalization of LSTM for tree-structured network topologies, which allows to process trees with arbitrary branching factors.

The first model to make use of RNN for a RE task was proposed by Liu et al. (2015). The authors introduced a CNN-based model applied on the shortest dependency path, augmented with a RNN-based feature designed to model subtrees attached to the shortest path. Miwa & Bansal (2016) introduced a variant of the TreeLSTM used to compute bidirectional (bottom-up and top-down) tree representations for performing relationship classification. Their model uses different weight matrices depending on whether a node belong to the shortest path or not.

In this paper, we use two deep-learning strategies to address the problem of RE. The first one is a MultiChannel Convolutional Neural Network (MCCNN) introduced in Quan et al. (2016) for biomedical RE. Inspired by the three-channel RGB image processing models, it consider different embedding channels (*i.e.*, different word embeddings versions for each word), allowing to capture different aspects of input words. The second model we used is the TreeLSTM model described in Tai et al. (2015) and more specifically its Child-Sum version. This model is suitable for processing dependency trees since it handles trees with arbitrary branching factors and no order between children of a node.

### 2.2 MULTI-TASK LEARNING

Machine learning methods and particularly deep learning ones usually require lots of annotated data in order to obtain reasonable performances. For certain tasks that does not require expert knowledge, such as the recognition of simple objects in an image, gathering lots of annotated data is relatively, easy using for instance crowd-sourcing. Some tasks, such as recognizing a relationship between complex entities that is mentioned in a biomedical scientific publication, are more complex, and the obtention of large corpora in this case can be expensive and time consuming. Several methods have been explored to deal with the lack of training data, such as *bootstrapping* (Jones et al., 1999), which allows accurate training from a small amount of labeled data, along with a large amount of unlabeled data; or *self-training* approaches McClosky et al. (2006) that artificially augment the labeled training set with examples from unlabeled datasets, using labels predicted by the model itself.

Beside, several studies have focused on transferring knowledge acquired from related tasks to help perform a new related task. For instance, Fei-fei et al. (2006) proposed a Bayesian approach to perform *one shot* learning, (*i.e.*, learning to categorize objects from a single example) that takes advantage of knowledge coming from previously learned categories.

*Multi-task Learning* is a learning approach in which performances on a given task are improved using information contained in the training signals of auxiliary related tasks Caruana (1997). It is a form of inductive transfer where the auxiliary task introduce an inductive bias during training. This is usually done by training tasks in parallel while using a shared representation (Sutton et al., 2007; Ando & Zhang, 2005). In Collobert et al. (2011), the authors jointly trained a CNN on various natural language processing tasks including part-of-speech tagging, chunking, named entity recognition, and semantic role labeling. They showed that sharing a portion of the network weights during training led to better performances for all the individual tasks.

## 3 MODELS

We consider in this article MCCNN and TreeLSTM models that both compute a fixed-size vector representation for the whole sentence by composing input embeddings. A score is then computed for each possible type or relationship (*e.g.*, negative, positive or speculative) between two identified entities. The number of possible relationship types depends on the task (see Section 4).

In this section, we first introduce the embedding input layer, which in common to both approaches (*i.e.*, MCCNN and TreeLSTM); Then, we detail how each approach composes sequences of embedding in order to compute an unique vectorial sentence representation; Finally, we present the scoring layer, which is common to both approaches.

### 3.1 INPUT LAYER

Both models are fed with *word embeddings* (*i.e.*, continuous vectors) of dimension $d_w$, along with extra *entity embeddings* of size $d_e$, which are concatenated to word embeddings. Formally, given a sentence of $N$ words, $w_1, w_2, \ldots, w_N$, each word $w_i \in W$ is first embedded in a $d_w$-dimensional vector space by applying a lookup-table operation:

$$LT_W(w_i) = W_{w_i} \ ,$$

where the matrix $W \in R^{d_w \times |W|}$ represents the parameters to be trained in this lookup layer. Each column $W_{w_i} \in R^{d_w}$ corresponds to the vector embedding of the $w_i^{\ th}$ word in our dictionary $W$.

Three entity embeddings (coming from a simple 3-elements dictionary) enable to distinguish between words which compose either the first entity, the second entity or are not part of any entity in a sentence. They are respectively called *first entity*, *second entity* and *other* embeddings. Finally, word and entity embeddings are concatenated to form the input corresponding to a given word. Let's denote $x_i$ the concatenated input corresponding to the $i^{th}$ word.

### 3.2 COMPOSITION LAYERS

Both models take the embeddings as input and output a fixed-size representation $r_s$ of size $d_s$. This section details the two models used in this study.

#### 3.2.1 MCCNN

The MCCNN models applies a variable kernel size CNN to multiple input channels of word embeddings. More formally, given an input sequence $x_1, \ldots, x_N$, applying a kernel of size $k$ to the $i^{th}$ window is done using the following formula:

$$C = h(\sum_{j=1}^{c} W[x_{\frac{i-1}{2}}, \ldots, x_i, \ldots, x_{\frac{i+1}{2}}]^j + b)$$

where $[\ ]^j$ denotes the concatenation of inputs from channel $j$, $W \in \mathcal{R}^{(d_w+d_e) \times d_h}$ and $b \in \mathcal{R}^{d_h}$ are the parameters , $h$ is a pointwise non-linear function such as the hyperbolic tangent and $c$ is the number of input channels. Inputs with indices exceeding the input boundaries ($\frac{i-1}{2} < 1$ or $\frac{i+1}{2} > N$)

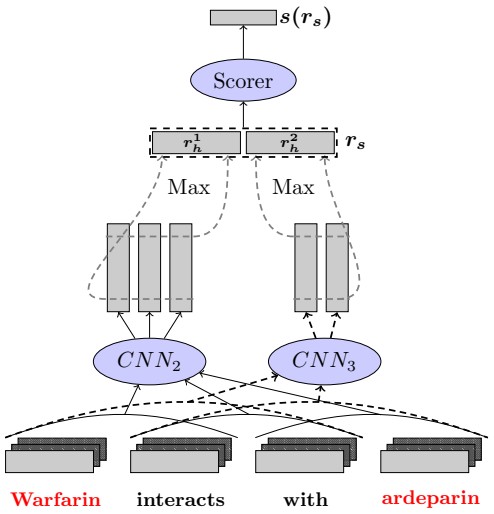

Figure 2: Visualization of the MCCNN model with three channels and two CNN with kernels of size 2 and 3 respectively. $CNN_i$ denotes a CNN with a kernel size $i$. Red words correspond to the entities.

Figure 3: Visualization of the TreeLSTM model. Red words correspond to the entities.

are mapped to a special padding vector (which is also learned). A fixed size representation $r_h \in \mathcal{R}^{d_h}$ is then obtain by applying a max-pooling over time:

$$r_h = \max C$$

We denote $K$ the number of kernel with different sizes. A sentence representation $r_s \in \mathcal{R}^{d_s}$ (with $d_s = K * d_h$) is finally obtained by concatenating the output corresponding to the $K$ kernels

$$r_s = [r_h^1, \ldots, r_h^k] \,,$$

where $r_h^k$ correspond to the output of the $k^{th}$ kernel. Figure 2 illustrates the structure of a two-channel CNN, with two kernels of size 2 and 3, on a four-words sentence.

### 3.2.2 TREELSTM

The TreeLSTM model (Tai et al., 2015) processes the dependency tree associated with an input sentence in a bottom-up manner. This is done by recursively processing the nodes of the tree, using their child representations as input. The transition function for a node $j$ and a set of children $C(j)$ is given by the following set of equations:

$$\tilde{h}_t = \sum_{k \in C(j)} h_k$$
$$i_j = \sigma(W^{(i)} x_j + U^{(i)} \tilde{h}_j + b^{(i)})$$
$$f_{jk} = \sigma(W^{(f)} x_j + U^{(f)} h_k + b^{(f)})$$

$$o_j = \sigma(W^{(o)} x_j + U^{(o)} \tilde{h}_j + b^{(o)})$$
$$u_j = \tanh(W^{(u)} x_j + U^{(u)} \tilde{h}_j + b^{(u)})$$
$$c_j = i_j \odot u_j + \sum_{k \in C(j)} f_{jk} \odot c_k$$
$$h_j = o_j \odot \tanh(c_j),$$

where $\sigma$ denotes the logistic function, $\odot$ the element-wise multiplication, $x_j \in \mathcal{R}^{d_w + d_e}$ is the input for node $j$, $h_k \in \mathcal{R}^{d_h}$ is the hidden state of the $k^{th}$ child. Each TreeLSTM unit is a collection of vectors: an input gate $i_j$, a forget gate $f_{jk}$, an output gate $o_j$, a memory cell $c_j$ and and hidden state $h_j$. The matrices $W$ and $U$ and the vectors $b$ are the weight and bias parameters to train. The

TreeLSTM outputs a sentence representation $r_s \in \mathcal{R}^{d_s}$ corresponding to the output state $o_j$ of the top tree node (*i.e.*, the *root* node of the dependency tree that spans all the others). Figure 3 illustrates the structure of the TreeLSTM computed for a four-words sentence.

## 3.3 SCORING LAYER

Both the MCCNN and the TreeLSTM models output an unique vector representation $r_s \in \mathcal{R}^{d_s}$ that takes the entire sentence into account. This representation is used to feed a single layer neural network classifier, which outputs a score vector with one score for each possible type of relationships. This vector of scores is obtained using the following formula:

$$s(r_s) = W^{(s)} r_s + b^{(s)} \ ,$$

where $W^{(s)} \in \mathcal{R}^{d_s \times |S|}$ and $b^{(s)} \in \mathcal{R}^{|S|}$ are the trained parameters of the scorer, $|S|$ is the number of possible relations. The scores are interpreted as probabilities using a softmax layer (Bishop, 2006).

## 4 DATASETS

We explore how RE tasks that focus on a type of relationship associated with scarce resources may take advantage from existing corpora, in other words how completing a small training corpus with a larger one may help the RE task when the latter is annotated with a different type of relationships. For this purpose, we selected *(i)* two small biomedical corpora, SNPPhenA and the EU-ADR corpus and *(ii)* three larger corpora, the SemEval 2013 DDI corpus, the ADE corpus and the reACE corpus. These corpora are publicly available and detailed in the following section. Table 4.2 summarizes the main characteristics of these five corpora and the following section details them.

### 4.1 SMALL CORPORA

- **SNPPhenA** (Bokharaeian et al., 2017) is a corpus of abstracts of biomedical publications, obtained from PubMed[1], annotated with two types of entities: *single nucleotide polymorphisms* (SNPs) and *phenotypes*. Relationships between these entities are annotated and classified in 3 categories: *positive*, *negative* and *neutral* relationships. The *neutral* relationship type is used when no relationship is mentioned in the sentence between two annotated entities.

- **EU-ADR** (van Mulligen et al., 2012) is a corpus of abstracts obtained from PubMed and annotated with *drugs*, *disorders* and drug targets (*proteins/genes* or *gene variants*) entities. It is composed of 3 subcorpora, focusing either on target-disease, target-drug or drug-disease relationships. Each of them consist of 100 abstracts. Annotated relationships are classified in 3 categories: *positive*, *speculative* and *negative associations* (PA, SA and NA respectively). In Bravo et al. (2015), performances are assessed over the TRUE class, which is composed of the classes PA, SA and NA, in contrast with the FALSE class composed of sentences where two entities are co-occurring, but without relationship annotated between them.

### 4.2 LARGE CORPORA

- **SemEval 2013 DDI** (Herrero-Zazo et al., 2013) consists of texts from DrugBank and MEDLINE and is annotated with drugs. Drug mentions are categorized in several types: *drug*, *brand*, *group* and *drug_n* (*i.e.*, active substances not approved for human use). Relationships between two drug mentions are annotated and classified in 4 categories: *mechanism*, *effect*, *advice* and *int*. *int* is the broader and default category for DDI, when no more detail can be provided.

- **ADE-EXT** (Adverse Drug Effect corpus, extended) (Gurulingappa et al., 2012) consists of MEDLINE case reports, annotated with *drug* and *conditions* (*e.g.*, diseases, signs and symptoms) along with untyped relationships between them, when one is mentioned.

---

[1]https://www.ncbi.nlm.nih.gov/pubmed/

- **reACE** (Edinburgh Regularized Automatic Content Extraction) (Hachey et al., 2012) consists of English broadcast news and newswire annotated with *organization*, *person*, *fvw* (facility, vehicle or weapon) and *gpl* (geographical, political or location) entities along with relationships between them. Relationships are classified in five categories (*general-affiliation*, *organisation-affiliation*, *part-whole*, *personal-social* and *agent-artifact*).

| Corpus | Subcorpus | Train Size | | Test Size | | #Entity Types | #Relation Types |
|---|---|---|---|---|---|---|---|
| | | sent. | rel. | sent. | rel. | | |
| SNPPhenA | – | 362 | 935 | 121 | 365 | 2 | 3 |
| EU-ADR | drug-disease | 244 | 176 | – | – | 4 | 3 |
| | drug-target | 247 | 310 | | | 4 | 3 |
| | target-disease | 355 | 262 | | | 4 | 3 |
| SemEval | DrugBank | 5,675 | 3,805 | 973 | 889 | 4 | 4 |
| 2013 DDI | MEDLINE | 1,301 | 232 | 326 | 95 | 4 | 4 |
| ADE-EXT | – | 5,939 | 6,701 | – | – | 2 | 1 |
| reACE | – | 5,984 | 2,486 | – | – | 4 | 5 |

Table 1: Main characteristics of the corpora. Two corpora are divided in subcorpora. The sizes of the training and test corpora are reported in term of number of sentences (sent.) and relationships (rel.). EU-ADR, ADR-EXT and reACE have no proper test corpus.

## 5 EXPERIMENTS

### 5.1 TRAINING AND EXPERIMENTAL SETTINGS

Our models are trained by minimizing a log-likelihood function over the training data. All parameters, including weights, biases and embeddings were updated via Backpropagation for the MCCNN and Backpropagation through Structure (BPTS) (Goller & Kuchler, 1996) for the TreeLSTM.

All the hyper-parameters were tuned using a 10 fold cross-validation by selecting the values leading to the best averaged performance, and fixed for the rest of the experiments. Word embeddings were pre-trained PubMed abstracts using the method described in Lebret & Collobert (2013). These abstracts correspond to all the abstracts published between January 1, 2014 and December 31, 2016, and available on Pubmed (around 3.4 million).

**MCCNN model.** Following Kim (2014) both channels are initialized with pre-trained word embeddings but gradients were back-propagated only through one of the channels. Hyper-parameters were fixed to $d_w = 100$, $d_e = 10$, $d_h = 100$ and $d_s = 200$. We applied a dropout regularization after the embedding layers.

**TreeLSTM model.** Dependency trees were obtained using the Stanford Parser (Chen & Manning, 2014). Hyper-parameters were fixed to $d_w = 100$, $d_e = 10$, $d_h = 200$ and $d_s = 200$. We applied a dropout regularization (Srivastava et al., 2014) after every TreeLSTM unit and after the embedding layers. The drop probability for each connexion was fixed to 0.25. All the parameters are initialized randomly except the word embeddings.

We evaluated performances in terms of precision (P), recall (R) and f-measure (F). For multi-label classifications, we report the macro-average performance[2]. Because no proper test corpus is provided with EU-ADR, we performed a 10 fold cross-validation using 10% of the corpus for the validation and 10% for the test of our models. For SNPPhenA, we performed a cross-validation using 10% of the corpus for the validation and the provided test corpus for testing.

### 5.2 CROSS-CORPUS STUDY

In this subsection, we present our cross-corpus training strategy and its results. For each fold of our cross-corpus experiments, the same network, initialized with random weight, is used for the different corpora (*i.e.*, same embedding layer and TreeLSTM weights), except for the scorer, which is

---

[2]The macro-average score is less impacted by the performance for classes whith very few test samples (and thus a high variance). For that reason, this score is more representatative of the performance of our model.

different for each corpus as the number and types of relationships may change. During the training phase, we randomly pick training sentences from the mixed corpora. Table 2 presents the results of the cross-corpus study. For each of the 10 folds, we performed 10 experiments starting from different random weight initializations. Thus, each result is an average of 100 experiments. We observe that for the TreeLSTM model, additional data consistently improved the performances. More interestingly, this phenomenon occurred even for corpora with different types of entities such as the combination of SNPPhenA and SemEval 2013 DDI and, to a lesser extend, for a corpus outside of the biomedical domain (reACE). This phenomenon was not observed for the MCCNN model for which performance tended to decrease slightly when using the cross-corpus learning strategy.

| Test Corpus | Model | Train corpus | P | R | F | $\sigma_F$ |
|---|---|---|---|---|---|---|
| **SNPPhenA** | TreeLSTM | SNPPhenA | 58.9 | 73.8 | 65.5 | 0.041 |
| | | + SemEval 2013 DDI | 65.2 | 71.1 | **68.0** | 0.047 |
| | | + ADE-EXT | 62.8 | 72.1 | 67.2 | 0.034 |
| | | + reACE | 61.8 | 74.3 | 67.1 | 0.036 |
| | MCCNN | SNPPhenA | 55.1 | 75.0 | **63.3** | 0.048 |
| | | + SemEval 2013 DDI | 55.3 | 74.4 | 63.3 | 0.049 |
| | | + ADE-EXT | 56.1 | 73.2 | 63.2 | 0.048 |
| | | + reACE | 53.2 | 70.9 | 60.6 | 0.041 |
| **EU-ADR drug-disease** | TreeLSTM | EU-ADR drug-disease | 74.8 | 84.1 | 79.1 | 0.123 |
| | | + SemEval 2013 DDI | 74.8 | 90.6 | **82.0** | 0.131 |
| | | + ADE-EXT | 73.9 | 88.2 | 80.4 | 0.137 |
| | | + reACE | 74.3 | 91.1 | 79.3 | 0.143 |
| | MCCNN | EU-ADR drug-disease | 73.3 | 94.7 | **80.2** | 0.142 |
| | | + SemEval 2013 DDI | 72.6 | 87.9 | 76.6 | 0.143 |
| | | + ADE-EXT | 73.0 | 85.5 | 76.0 | 0.145 |
| | | + reACE | 74.1 | 91.5 | 79.2 | 0.138 |
| **EU-ADR drug-target** | TreeLSTM | EU-ADR drug-target | 72.4 | 90.6 | 80.2 | 0.109 |
| | | + SemEval 2013 DDI | 71.9 | 95.5 | **82.5** | 0.085 |
| | | + ADE-EXT | 70.2 | 96.7 | 80.9 | 0.092 |
| | | + reACE | 70.4 | 96.5 | 80.8 | 0.093 |
| | MCCNN | EU-ADR drug-target | 74.5 | 92.3 | **81.0** | 0.093 |
| | | + SemEval 2013 DDI | 74.9 | 88.8 | 80.0 | 0.106 |
| | | + ADE-EXT | 76.3 | 87.4 | 80.3 | 0.101 |
| | | + reACE | 73.4 | 92.1 | 80.5 | 0.078 |
| **EU-ADR target-disease** | TreeLSTM | EU-ADR target-disease | 77.0 | 89.7 | 82.7 | 0.064 |
| | | + SemEval 2013 DDI | 77.4 | 91.6 | **83.9** | 0.082 |
| | | + ADE-EXT | 77.7 | 89.5 | 83.3 | 0.069 |
| | | + reACE | 75.9 | 91.7 | 83.0 | 0.077 |
| | MCCNN | EU-ADR target-disease | 76.9 | 91.8 | **82.6** | 0.077 |
| | | + SemEval 2013 DDI | 77.6 | 90.6 | 82.5 | 0.071 |
| | | + ADE-EXT | 75.5 | 87.4 | 81.8 | 0.101 |
| | | + reACE | 77.1 | 91.2 | 82.0 | 0.068 |

Table 2: Impact of cross-corpus training in terms of precision (P), recall (R) and f-measure (F). $\sigma_F$ is the standard deviation of the f1-score.

## 5.3 COMPARISON WITH THE STATE OF THE ART

Table 3 presents a comparison of performances obtained with our approach *versus* two state-of-the-art systems applied to the RE tasks associated respectively with SNPPhenA and EU-ADR, respectively reported in Bokharaeian et al. (2017) and Bravo et al. (2015). Our results are obtained using, for each fold, an ensemble of the 5 best models (according to the validation) starting from different random initialization. The ensembling was done by averaging the scores $s(r_s)$ of each individual model, following Legrand & Collobert (2014). We report the 10 folds average performance. Both state-of-the-art systems use a combination of a shallow linguistic kernel with a kernel that exploits deep syntactic features. Our approach outperforms the performances reported for SNPPhenA and for the one EU-ADR subtasks and led to similar performances for the two remaining EU-ADR subtasks.

| Test corpus | Train corpus | P | R | F |
|---|---|---|---|---|
| SNPPhenA | **Bokharaeian et al. (2017)** SNPPhenA | 56.6 | 59.8 | 58.2 |
| | **This work** SNPPhenA + ADE-EXT | 64.5 | 75.2 | **69.4** |
| EU-ADR drug-disease | **Bravo et al. (2015)** EU-ADR drug-disease | 70.2 | 93.2 | 79.3 |
| | **This work** EU-ADR drug-disease + SemEval 2013 DDI | 74.8 | 90.6 | **82.0** |
| EU-ADR drug-target | **Bravo et al. (2015)** EU-ADR drug-target | 74.2 | 97.4 | 83.0 |
| | **This work** EU-ADR drug-target + SemEval 2013 DDI | 73.5 | 95.6 | **83.1** |
| EU-ADR target-disease | **Bravo et al. (2015)** EU-ADR target-disease | 75.1 | 97.7 | **84.6** |
| | **This work** EU-ADR target-disease + SemEval 2013 DDI | 78.7 | 91.4 | **84.6** |

Table 3: Performance comparison with the state of the art in terms of precision (P), recall (R) and f-measure (F), using ensembles of 5 models.

## 6 DISCUSSION

Results presented in Table 2 show that, in our settings, the TreeLSTM model benefits from a cross-corpus learning strategy, while it is useless, or sometimes counterproductive for the MCCNN model. One may think that the TreeLSTM model, due to its ability to exploit the syntactic structure of the sentence, is better at understanding the sentences from the small datasets by exploiting the syntactic patterns observed in the additional data. This idea is reinforced by the fact that even a corpus that does not share the same entities nor a close vocabulary, such as reACE in which no biomedical vocabulary appear, can be helpful for biomedical RE. This assessment could be interestingly explored in further work.

Surprisingly, the best results where consistently obtained using the SemEval 2013 DDI corpus as additional data, even for RE tasks that doesn't involve drugs like EU-ADR target-disease. Likewise, one might have thought that the ADE-EXT corpus could have been more suitable for the EU-ADR drug-disease corpus, since it shares common entities. Several ideas should be explored to better understand this phenomenon, such as the differences of relation and entity types between the different corpora, as well as the differences of types of texts in sources (*e.g.*, medical case report for ADE-EXT, news for reACE, research articles for the others). Higher level syntactic analysis (such as the average distance between the two entities or the nature of the lowest common ancestor in the dependency graph) could provide insights on this question, and help in characterizing the right corpus to select for a cross-corpus training.

For the TreeLSTM model, we also tried to train models with multiple additional corpora but did not obtained better performances. For each of the 4 RE tasks studied, the results were consistently on par with the performances obtained using only the additional corpus leading the worst cross-corpus performances. Further work should be done to better understand this phenomenon.

Finally, it would be interesting to enrich our model with additional feature such as POS or morpho-syntactic ones. More sophisticated TreeLSTM model, taking the dependency tags into account, in addition to the dependency structure, would also be worth exploring.

## 7 CONCLUSION

In this paper, we empirically demonstrated that a cross-corpus learning strategy can be beneficial to tackle biomedical RE tasks for which few annotated resources are available, when using the TreeLSTM model. Interestingly, we showed that any additional corpus, even when focusing on unrelated domain can carry useful information and lead to improved performances. Additionally, the cross-corpus approach led to the best published results for 2 biomedical RE task focusing on SNP-phenotype and drug-disease and to state-of-the-art result for two others focusing on target-disease and target-drug. We think that cross-corpus training could be reproduced and thus valuable in other specialized domains in which training resources are scarce.

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
