# OpenReview forum: "Cross-Corpus Training with TreeLSTM for the Extraction of Biomedical Relationships from Text"
_ICLR.cc/2018/Conference — Invite to Workshop Track_

### Official Review · AnonReviewer1 · 2017-11-24

**Rating:** 4
**Confidence:** 4

**Review:**

SUMMARY.

The paper presents a cross-corpus approach for relation extraction from text.
The main idea is complementing small training data for relation extraction with training data with different relation types.
The model is also connected with multitask learning approaches where the encoder for the input is the same but the output layer is different for each task. In this work, the output/softmax layer is different for each data type, while the encoder is shared.
The authors tried two different sentence encoders (cnn-based and tree-lstm), and final results are calculated on the low resource dataset.

Experimental results show that the tree-rnn encoder is able to capture valuable information from auxiliary data, while the cnn based does not.

----------

OVERALL JUDGMENT
The paper shows an interesting approach to data augmentation with data of different type for relation extraction.
I would have appreciated a section where the authors explain briefly what relation extraction is maybe with an example.
The paper is overall clear, although the experimental section has to be improved I believe.
From section 5.2 I am not able to understand the experimental setting the authors used, is it 10-fold CV? Did the authors tune the hyperparameters for each fold?
Are the results in table 3 obtained with tree-lstm?
What kind of ensembling did the authors chose for those experiments?
The author overstates that their model outperforms the state-of-the-art models they compare to, but that is not true for the EU-ADR dataset where in 2 out of 3 relation types the proposed model performs on par with the state-of-the-art model.
Finally, the authors used only one auxiliary dataset at the time, it would be interesting to see whether using all the auxiliary dataset together would improve results even more.

I would suggest the author also to check and revise citations (CNN's are not Collobert et al. invention, the same thing for the maximum likelihood objective) and more in general to improve the reference on relation extraction literature.

---

### Official Review · AnonReviewer3 · 2017-11-28
**This paper describes cross-corpus studies using Tree-LSTM and MCCNN-based RE models in the biomedical domain. Experimantal results show that some combinations of different corpora lead to better performance.**

**Rating:** 5
**Confidence:** 4

**Review:**

This is a well-written paper with sound experiments. However, the research outcome is not very surprising.

- Only macro-average F-scores are reported. Please present micro-average scores as well.
- The detailed procedure of relation extraction should be described. How do you use entity type information? (Probably, you did not use entity types.)
- Table 3: The SotA score of EU-ATR target-disease (i.e. 84.6) should be in bold face.
- Section 5.3: Your system scorers in Table 3 are not consistent with Table 2 scores.
- Page 8. "Our approach outperforms ..." The improvement is clear only for SNPPhenA and EU-ADR durg-disease.

Minor comments:

- TreeLSTM --> Tree-LSTM
- Page 7. connexion --> connection
- Page 8. four EU-ADR subtasks --> three ...
 - I suggest to conduct transfer learning studies in the similar settings.

---

### Official Review · AnonReviewer2 · 2017-12-04
**Lack of Novelty**

**Rating:** 3
**Confidence:** 5

**Review:**

This paper proposes to use Cross-Corpus training for biomedical relationship extraction from text.

- Many wording issues, like citation formats, grammar mistakes, missing words,
  e.g., Page 2: it as been

- The description of the methods should be improved.
  For instance, why the input has only two entities? In many biomedical sentences, there are more than two entities. How can the proposed two models handle these cases?

- The paper just presents to train on a larger labeled corpus and test on a task with a smaller labeled set. Why is this novel?
  Nothing is novel in the deep models (CNN and TreeLSTM).

- Missing refs, like:
 A simple neural network module for relational reasoning, Arxiv 2017

---

### Decision · Program_Chairs · 2018-01-29
**ICLR 2018 Conference Acceptance Decision**

**Decision:**

Invite to Workshop Track

**Comment:**

We encourage the authors to improve the mentioned aspects of their work in the reviews.